# GDrag: Towards General-Purpose Interactive Editing with Anti-ambiguity Point Diffusion

**Xiaojian Lin**[1*]**, Hanhui Li**[1,2*]**, Yuhao Cheng**[3]**, Yiqiang Yan**[3]**, Xiaodan Liang**[1,2,4†]
[1]Shenzhen Campus of Sun Yat-Sen University
[2]Guangdong Key Laboratory of Big Data Analysis and Processing [3]Lenovo Research
[4]Peng Cheng Laboratory
`linxj68@mail2.sysu.edu.cn lihh77@mail.sysu.edu.cn`
`chengyh5@lenovo.com yanyq@lenovo.com xdliang328@gmail.com`

## Abstract

Recent interactive point-based image manipulation methods have gained considerable attention for being user-friendly. However, these methods still face two types of ambiguity issues that can lead to unsatisfactory outcomes, namely, intention ambiguity which misinterprets the purposes of users, and content ambiguity where target image areas are distorted by distracting elements. To address these issues and achieve general-purpose manipulations, we propose a novel task-aware, training-free framework called GDrag. Specifically, GDrag defines a taxonomy of atomic manipulations, which can be parameterized and combined unitedly to represent complex manipulations, thereby reducing intention ambiguity. Furthermore, GDrag introduces two strategies to mitigate content ambiguity, including an anti-ambiguity dense trajectory calculation method (ADT) and a self-adaptive motion supervision method (SMS). Given an atomic manipulation, ADT converts the sparse user-defined handle points into a dense point set by selecting their semantic and geometric neighbors, and calculates the trajectory of the point set. Unlike previous motion supervision methods relying on a single global scale for low-rank adaption, SMS jointly optimizes point-wise adaption scales and latent feature biases. These two methods allow us to model fine-grained target contexts and generate precise trajectories. As a result, GDrag consistently produces precise and appealing results in different editing tasks. Extensive experiments on the challenging DragBench dataset demonstrate that GDrag outperforms state-of-the-art methods significantly. The code of GDrag is available at `https://github.com/DaDaY-coder/GDrag`.

## 1 Introduction

With the recent advances in generative models (Rombach et al., 2022), many notable image editing methods (Zhang et al., 2023b; Brooks et al., 2023; Kawar et al., 2023; Chen et al., 2023) have emerged, enabling users to generate or edit image content effectively. Among these methods, a new paradigm that edits images by interactively dragging points on images (Pan et al., 2023b), provides an intuitive and convenient way to obtain desirable images. Specifically, to edit an image, a user places several points on the image and drags them to target positions, as shown in Figure 1. These points are considered as handle points, and our task is to move image content along the trajectories of handle points to seamlessly blend the dragged content into its context. To this end, methods of this type alternatively conduct two key processes in iterations, i.e., motion supervision that modifies image content guided by tailored optimization objectives, and point tracking that adjusts the dragging trajectories by calculating intermediate positions of handle points.

---

[*]Both authors contributed equally.
[†]Corresponding author.

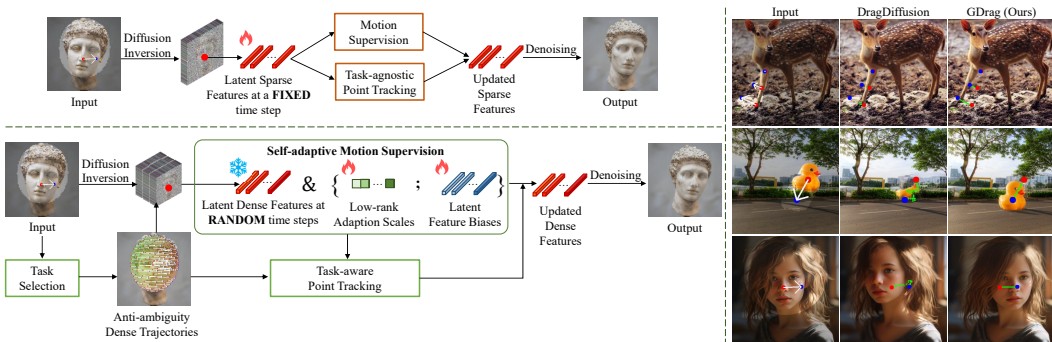

Figure 1: Paradigm comparison between previous dragging methods (left-top) and the proposed GDrag method (left-bottom). GDrag estimates task-aware dense trajectories and adopts more fine-grained motion supervision. Hence, GDrag obtains smoother, more reasonable trajectories (labeled in green) and appealing results.

Although current point-based diffusion methods (Shi et al., 2024b; Mou et al., 2024; Ling et al., 2024) already achieve relatively good editing effects, they still face certain limitations. Specifically, as existing methods model various editing tasks implicitly, they suffer from two types of ambiguities, i.e., **intention ambiguity** that mixes multiple possible editing tasks into a single trajectory, and **content ambiguity** that fails to identity and preserve the targets. For example, DragDiffusion (Shi et al., 2024b) updates the positions of handle points after each optimization step regardless of the task, which tends to yield less reasonable and drifted trajectories. FreeDrag (Ling et al., 2024) alleviates this issue by constraining the trajectories to straight lines. However, such 2D trajectories cannot fully represent 3D manipulations like out-of-plane rotations. Regarding content ambiguity, most existing methods rely on a user-specific denoising time step to select latent features of handle points for optimization. However, the information encoded in latent features at different time steps varies significantly. Since different editing tasks require different levels of information, it is impractical to generate satisfying results by exploiting latents only at one time step.

To address the above ambiguity issues in current methods, we propose to model editing tasks explicitly in this paper. Intuitively, we can divide a complex manipulation into multiple basic tasks, each with intents that are clear and easy to perform. This paradigm not only reduces the difficulty of editing but also provides an opportunity to optimize edited images accordingly. For example, in a task that involves simply moving a target to another position, we can encourage the model to preserve the target's features as much as possible. Conversely, for tasks like target removal, it is more reasonable to update the target features rapidly.

Therefore, we propose a novel task-aware, optimization-based framework for general-purpose interactive editing, named GDrag. GDrag categorizes point-based image manipulations into three atomic tasks: relocation, rotation (both in-plane and out-of-plane), and non-rigid transformation (e.g., scaling, content creation/removal). Specifically, given an input image, GDrag first requires the user to specify the task through simple interactions (e.g., a single click). We then introduce an anti-ambiguity dense trajectory estimation method (ADT), which addresses the lack of contextual information in the sparse handle points selected by users. Based on the specific task, ADT selects the semantic and geometric neighbors of handle points as the context and calculates their dense corresponding trajectories. As we will demonstrate in the following sections, our dense trajectories alleviate the ill-posedness of 2D lines in representing 3D deformations, thereby providing more reliable guidance for subsequent optimizations. Additionally, we introduce a self-adaptive motion supervision method (SMS) to optimize latent features based on their dense trajectories. SMS differs from previous motion supervision methods in two key aspects: First, it aims to optimize biases for latent features randomly sampled from all denoising time steps, better exploiting the generative capabilities of diffusion models. Second, it optimizes scaling maps that control the effects of low-rank adaptation (Hu et al., 2021). Thus, it can preserve target contents at different granularity levels and alleviate content ambiguities. Consequently, as shown in Figure 1, GDrag can achieve more reasonable trajectories and higher-quality images compared to conventional methods.

Our contributions are summarized as follows:

• We propose GDrag, the first optimization-based framework that explicitly models editing tasks to handle ambiguities.

• Based on different task types, we propose the ADT method to construct a dense dragging point set and calculate their trajectories, which can offer more comprehensive and reasonable prior knowledge for image editing.

• We propose the SMS method that introduces task-aware, fine-grained optimization parameters to refine latent features. This allows us to address content ambiguities and improve the quality of edited images.

• By showcasing different experimental results, we demonstrate the powerful editing capabilities of GDrag, achieving superior performance on DragBench.

## 2 RELATED WORK

### 2.1 IMAGE EDITING

Image editing has always been a hot topic and widely applied in various fields. Previous methods based on generative adversarial networks (GANs) (Goodfellow et al., 2014; Karras et al., 2020) already achieve considerable performance (Afifi et al., 2021; Parihar et al., 2022; Shen et al., 2020; Zhou et al., 2022). Recently, with the rapid development of large-scale diffusion models (Rombach et al., 2022), diffusion-based methods become the mainstream for image editing (Epstein et al., 2023; Mao et al., 2023; Hertz et al., 2022; Tumanyan et al., 2023). Especially, text-based image synthesis and editing (Brooks et al., 2023; Chang et al., 2023; Li et al., 2024a; Cheng et al., 2024; Gao et al., 2024; Ahn et al., 2024) that exploit the learned correspondences between texts and images from pretrained large multi-modal networks (Radford et al., 2021) provides a convenient scheme for related tasks. For instance, Brooks et al. (2023) propose an instruction-based method, in which editing intentions can be depicted by natural instructions. Autostudio (Cheng et al., 2024) introduces a framework that supports multi-turn text-based editing. However, as texts are high-level summaries of user intentions, they inevitably result in more ambiguities compared with interactive points.

### 2.2 INTERACTIVE POINT-BASED MANIPULATION

Interactive point-based image editing allows users to easily edit images by manipulating points in images. A pioneering method of this type is GragGAN (Pan et al., 2023a), which proposes the idea of motion supervision and point tracking. However, due to the utilization of GANs and optimizations on 1D latent features, the editing ability of GragGAN is limited in complex scenarios. With the recent success of diffusion models in image generation, the quality of synthesized images generated by dragging-based methods (Mou et al., 2024; Shi et al., 2024b; Ling et al., 2024; Liu et al., 2024; Zhang et al., 2024) has been improved notably. For example, DragonDiffusion (Mou et al., 2024) introduces a gradient-guidance-based energy function to model diffusion sampling, which can better perverse the characteristics of targets. A similar framework is explored in DragDiffusion (Shi et al., 2024b), which also proposes the DragBench dataset to facilitate related research. To mitigate handle point deviations, FreeDrag (Ling et al., 2024) constrains updates of point positions to straight lines to generate more stable editing results. DragNoise (Liu et al., 2024) utilizes feature maps from intermediate layers of the denoising network to alleviate gradient vanishing. GoodDrag (Zhang et al., 2024) proposes to alternate between drag and denoising steps to achieve finer controls.

Dragging-based methods can also benefit from diffusion models that can generate content beyond images. For instance, DragAnything (Wu et al., 2024), DragNUWA (Yin et al., 2023), and Motion-I2V (Shi et al., 2024a) exploit video generators to achieve point-based video editing. DragAPart (Li et al., 2024b) learns motion priors in parts of 3D objects for dragging. However, as we have emphasized, most current methods do not seek the explicit interpretation of user intentions, and hence they still suffer from intention and content ambiguity.

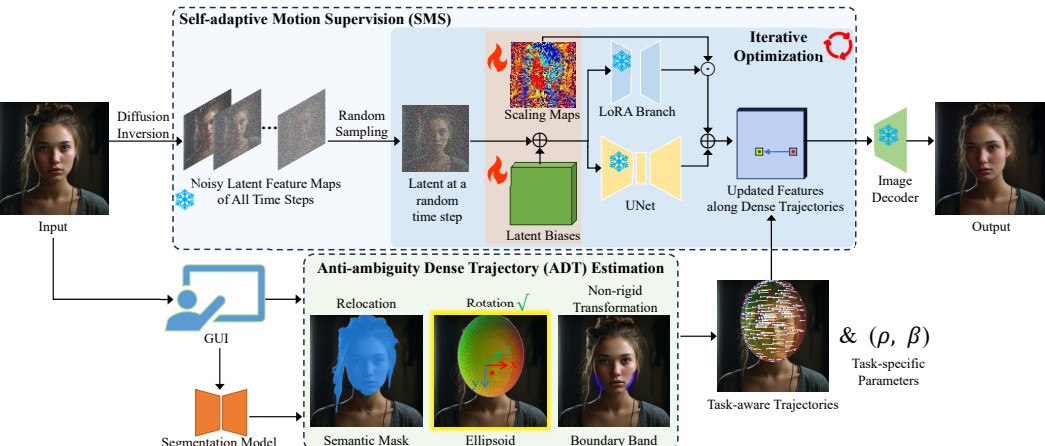

Figure 2: The proposed GDrag framework. The key idea of GDrag is to reduce intention and content ambiguities, which is accomplished by the proposed anti-ambiguity dense trajectory estimation method (ADT) and the self-adaptive motion supervision method (SMS). Given user-specific sparse handle points and the editing task, ADT selects a dense set of points that encode rich contextual information and estimates the corresponding task-aware trajectories. Utilizing these trajectories and a pair of task-specific parameters, the SMS method adjusts the positions and latent features of the dense points by optimizing latent feature biases and scaling maps, thereby achieving fine-grained editing results.

## 3 METHODOLOGY

In this section, we present the details of the proposed GDrag method. We start by introducing the overall framework of GDrag and formulating our task. Next, we present the two core components of GDrag, including a dense trajectory estimation method (ADT) to address intention ambiguities and a self-adaptive motion supervision method (SMS) to solve content ambiguities.

### 3.1 GDRAG FRAMEWORK

Conventional dragging-based methods, which lack explicit modeling of editing intentions, inevitably create ambiguities. These ambiguities not only increase the difficulty of editing but also reduce the quality of the edited images. Therefore, the key idea of our GDrag method is to define a set of atomic manipulation tasks with clear and explicit intentions. These atomic tasks should be parameterized and guide subsequent processing, enabling us to complete diverse editing tasks within a unified framework. To this end, we propose the GDrag framework as shown in Figure 2.

Specifically, we categorize point-based manipulations into three types, including *relocation*, *rotation*, and *non-rigid transformation*. Relocation involves translating the target. Rotation includes both in-plane and out-of-plane rotations, meaning the target rotates within or out of the image plane. Non-rigid transformation comprises isotropic and anisotropic scaling, as well as the removal or creation of content within the target. To obtain satisfying results, it is reasonable to perform manipulations with different

Table 1: Main notations within GDrag.

| Symbol | Definition |
|---|---|
| $L$ | Length of a trajectory. |
| $N$ | Number of optimizations after each movement. |
| $T$ | Number of denoising steps. |
| $\mathbf{z}_t$ | Latent features at the $t$-th denoising step. |
| $\hat{\mathbf{z}}_t^n$ | $\mathbf{z}_t$ after $n$ optimization steps. |
| $\mathbf{b}$ | Optimizable latent feature biases. |
| $\mathbf{s}$ | Optimizable low-rank adaption scales. |
| $\mathcal{M}$ | Target mask. |
| $\mathcal{P}^* \& \mathcal{G}$ | Dense points and their trajectories. |
| $\mathcal{U}, \mathcal{U}'$ | UNet and its low-rank side-branch for denoising. |
| $\rho, \beta$ | Task-conditioned parameters. |

degrees of freedom and calculate different point trajectories for these three tasks. For instance, in the relocation task, we should preserve the characteristic of the target as much as possible, while allowing flexible deformations in the non-rigid transformation task that may alter the target significantly.

Therefore, given a target image and user-specific trajectories of handle points, we assume these trajectories belong to an atomic task and manually specify its type. Note that complex manipulations can be divided into atomic tasks (as shown in Figure 6), so we focus on a single atomic task in the following sections. Since the trajectories of handle points are sparse and limited in representing the context during editing, we propose the ADT method to identify the semantic and geometric neighbors of dragging points to construct a dense point set and calculate their trajectories based on the task type. Unlike previous point-tracking methods (Pan et al., 2023b; Shi et al., 2024b; Liu et al., 2024) that update point positions by searching pre-defined neighboring areas, we move our dense points only on their own trajectories, which is feasible as their trajectories are less ambiguous. To generate the desired edited image, our SMS method optimizes the latent features of dense points extracted by a pretrained diffusion model during their movements. Besides, a pair of task-conditioned parameters are used in our optimization process to balance target preservation and editing flexibility.

**Discussion**. Similar to our method, a few concurrent approaches (Zhang et al., 2024; Cui et al., 2024) also mention intention ambiguities. However, they either conduct post-editing evaluations (Zhang et al., 2024) or utilize resource-intensive large language models (Cui et al., 2024). In contrast, our method is explicitly conditioned on tasks, which can be achieved through a few user-friendly interactions, such as a single click on option tabs in a graphical user interface, as demonstrated in Appendix A.9. Consequently, our method preserves the efficiency of point-based interactive editing while significantly reducing ambiguities.

## 3.2 ANTI-AMBIGUITY DENSE TRAJECTORY

The goal of our proposed ADT method is to obtain sufficient contextual information and reliable task-aware trajectories. Specifically, given $M$ user-specific handle points $\mathcal{P} = \{p_m = (x_m, y_m) | m = 0, ..., M - 1\}$ and their corresponding end positions $\mathcal{Q} = \{q_m = (x_m, y_m) | m = 0, ..., M - 1\}$, we first employ the off-the-shelf promptable segmentation model (Kirillov et al., 2023) to obtain the semantic segmentation mask of the target $\mathcal{M}$. We then construct the dense point set $\mathcal{P}^*$ and estimate point trajectories $\mathcal{G}$ based on the atomic task as follows:

**Relocation**. In the relocation task, points within the target move coherently with similar directions and distances. Therefore, we calculate $\mathcal{P}^*$ and $G$ by,

$$\mathcal{P}^* = \mathcal{M}, \quad \mathcal{G} = \{g_j^l = p_j + \Delta \frac{l}{L}\},$$

$$\Delta = \frac{1}{M} \sum_{m=1}^{M} q_m - p_m,$$

(1)

where $0 \leq j \leq |\mathcal{P}^*|$, $0 \leq l \leq L - 1$, and $L$ is the number of steps that move each point to its target. $\Delta$ denotes the mean displacement of handle points.

**Rotation**. Since recovering 3D rotations from 2D image-plane trajectories is ill-posed, we utilize an ellipsoid $\mathcal{E}$ as the alternative tool to calculate trajectories. Specifically, as shown in Figure 2, we first place $\mathcal{E}$ near the center of the target, of which the axis lengths and angles can be adjusted to coarsely cover the target. Let $\Phi(\mathcal{E})$ denote the orthographic projection of the surface points of $\mathcal{E}$ onto the image plane. We approximate $\mathcal{P}$ by finding their nearest neighbors from $\Phi(\mathcal{E})$ (denoted as $\mathcal{P}_{\mathcal{E}}$), and then rotate $\mathcal{E}$ around one of its three axes so that the projection of $\mathcal{P}_{\mathcal{E}}$ are close to $\mathcal{Q}$. Hereby, we obtain $\mathcal{P}^*$ and $G$ for the rotation task as follows,

$$\mathcal{P}^* = \mathcal{M} \cap \Phi(\mathcal{E}),$$

$$\mathcal{G} = \{g_j^l = \Phi(p_{\mathcal{E}, j}) + \Phi(d_j) \frac{l}{L}\},$$

(2)

where $d_j$ denotes the geographical displacement (i.e., on the ellipsoidal surface) of $p_{\mathcal{E}, j}$ after the rotation. Consequently, our generated trajectories can better represent 3D rotations compared to those calculated directly by handle points on the image plane. Moreover, our trajectories implicitly

encode the geometric information, such as varying distances of surface points of the target, which can guide subsequent optimizations.

**Non-rigid transformation**. To complete non-rigid transformations such as scaling, we focus on the boundary points, as they determine the shape and pose of the target. Particularly, we extract a boundary band from $\mathcal{M}$ with a fixed width of $r$. To minimize the influence of boundary points that are distant from $\mathcal{P}$ and achieve fine-grained, local transformations, we retain only boundary points that are among the $k$ nearest neighbors of any point in $\mathcal{P}$. Let $\Omega_k(\mathcal{M})$ denote the set of selected boundary points, we define $\mathcal{P}^*$ and $G$ as follows,

$$\mathcal{P}^* = \Omega_k(\mathcal{M}),$$
$$\mathcal{G} = \{g_j^l = p_j^* + (q_j^* - p_j^*)\frac{l}{L}\}, \tag{3}$$

where the starting and end positions of $p_j^*$ are set as those of its nearest handle point.

With the above definitions of task-aware dense trajectories, GDrag can clarify user intents. Figure 3 demonstrates the visual examples of our generated dense points and trajectories for different tasks. In the next section, we optimize the features of $\mathcal{P}^*$ along trajectories $\mathcal{G}$, thereby accomplishing various manipulations within a unified framework.

### 3.3 SELF-ADAPTIVE MOTION SUPERVISION

Motion supervision (Pan et al., 2023b) is the process that optimizes contextual features of handle points along their trajectories gradually. In current diffusion-based methods (Liu et al., 2024; Shi et al., 2024b; Ling et al., 2024), motion supervision requires a user-specific denoising time step to choose the latent features for optimization. However, recent studies (Choi et al., 2022; Croitoru et al., 2023) indicate that diffusion models tend to generate high-level content during the early denoising steps and finer details in the later stages. Therefore, optimizing latent features at a single time step is insufficient for producing desirable content. To address this issue, we introduce the SMS method in this section.

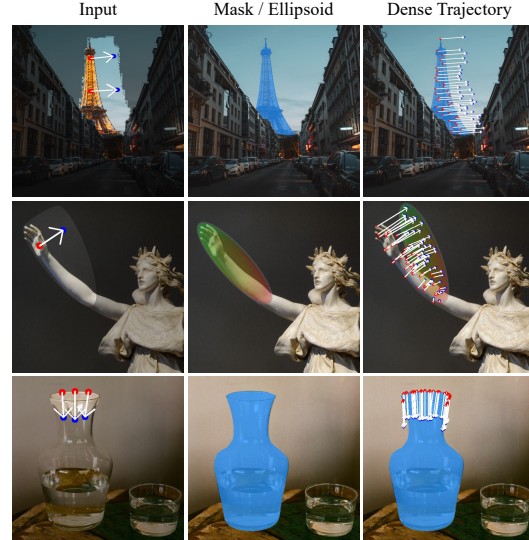

Figure 3: Visual examples of our generated dense trajectories for diverse tasks, including relocation (top), rotation (middle), and non-rigid transformation (bottom).

Following existing methods (Shi et al., 2024b; Mou et al., 2024), we employ DDIM inversion (Song et al., 2021) to transform the input image into a series of latent feature maps $\{\mathbf{z}_t\}$ within $T$ diffusion steps, where $0 \leq t \leq T - 1$ represents an arbitrary time step. $\mathbf{z}_0$ denotes the clean feature maps while $\mathbf{z}_{T-1}$ is the noise that can be easily sampled from certain distributions (e.g., $\mathcal{N}(\mathbf{0}, \mathbf{1})$). This diffusion process can be formulated as,

$$\mathbf{z}_t = \sqrt{a_t}\mathbf{z}_0 + \sqrt{1 - a_t}\epsilon, \tag{4}$$

where $\alpha_t$ is the Gaussian variance governing the transition probability of the Markov chain in DDIM and $\epsilon \sim \mathcal{N}(\mathbf{0}, \mathbf{1})$. To recover $\mathbf{z}_0$ and generate the edited image, we can alter a certain latent $\mathbf{z}_t$ and iteratively remove noise according to Eq. (4).

The original motion supervision method assumes that the latent feature maps of the desired edited image adhere to the same data priors as those of the input image. Therefore, it moves the handle points step-by-step and minimizes the difference between the corresponding features of the control point before and after a small movement. Given the dense trajectories $\mathcal{G}$ consisting of $L$ steps, if we naively run through all $T$ denoising time steps to exploit sufficient latent features, the computational complexity is $O(TL)$, which is a significant burden. Hence, we propose to perform optimizations $N$ times ($N \ll T$) with randomly selected time steps after each movement.

Specifically, let $0 \leq n \leq N - 1$ denote an arbitrary optimization step after a small movement on the dense trajectories. For the concise presentation, here we ignore the indexes of movement steps on the dense trajectories and denote the latent features at a randomly selected denoising time step $t$ as $\hat{\mathbf{z}}_t^n$. By substituting $\hat{\mathbf{z}}_t^n$ into Eq. (4), we can obtain the optimized latent feature maps $\hat{\mathbf{z}}_t^{n+1}$ as,

$$\hat{\mathbf{z}}_t^{n+1} = \sqrt{\alpha_t} \hat{\mathbf{z}}_0^{n+1} + \sqrt{1 - \alpha_t} \epsilon_t, \tag{5}$$

where $\epsilon_t$ is predicted by the pre-trained diffusion model. To calculate $\hat{\mathbf{z}}_0^{n+1}$, we first obtain $\hat{\mathbf{z}}_0^0$ from the unaltered latents as follows,

$$\hat{\mathbf{z}}_0^0 = \frac{1}{\sqrt{\alpha_{t+1}}} \Big( \mathbf{z}_{t+1} - \sqrt{1 - a_{t+1}} \epsilon_{t+1} \Big). \tag{6}$$

We then introduce the following self-adaptive scheme for updating $\hat{\mathbf{z}}_0^{n+1}$:

$$\hat{\mathbf{z}}_0^{n+1} = (1 - \rho) \hat{\mathbf{z}}_0^n + \rho \mathbf{b}^n. \tag{7}$$

Here, $\mathbf{b}^n$ denotes latent feature biases for optimization, which is initialized as $\mathbf{z}_0$ to ensure it follows the same data distribution of the original image. $\rho$ is a task-specific weighting factor, e.g., in the relocation task, a larger $\rho$ is preferred, as it generally leads to better preservation of the original latents.

Additionally, it is common practice to utilize low-rank adaption (LoRA) (Shi et al., 2024b; Ling et al., 2024; Liu et al., 2024), which fine-tunes the denoising UNet of the diffusion model with a low-rank side branch. As the bottleneck feature dimension (rank) of the LoRA branch is much smaller than that of the original diffusion model, fine-tuning this branch reduces the feature space of the model to be compactly around the target image, consequently helping to preserve the identity of the target. Specifically, let $\mathcal{U}$ and $\mathcal{U}'$ denote the original UNet and its fine-tuned LoRA branch, respectively. The outputs of $\mathcal{U}$ and $\mathcal{U}'$ are combined linearly with a single trade-off scalar, i.e., $\mathcal{U}(\hat{\mathbf{z}}_t^{n+1}) + s \cdot \mathcal{U}'(\hat{\mathbf{z}}_t^{n+1})$. Nevertheless, such a global combination is insufficient to model fine-grained and heterogeneous feature transformations. Hence, we also introduce an optimizable scaling map $\mathbf{s}^n$ in SMS as follows,

$$f(\hat{\mathbf{z}}_t^{n+1}) = \mathcal{U}(\hat{\mathbf{z}}_t^{n+1}) + \mathbf{s}^n \odot \mathcal{U}'(\hat{\mathbf{z}}_t^{n+1}), \tag{8}$$

where $\odot$ denotes the (broadcast) Hadamard product and $\mathbf{s}^n$ is of the same spatial resolution as $\hat{\mathbf{z}}_t^{n+1}$.

The desirable latents of the edited image can be obtained now by jointly optimizing $\mathbf{b}$ and $\mathbf{s}$ supervised by the following loss function,

$$\mathcal{L} = (1 - \beta) \mathcal{L}_{\text{align}} + \beta \mathcal{L}_{\text{smooth}} + \lambda \mathcal{L}_{\text{mask}}, \tag{9}$$

where the first loss term $\mathcal{L}_{\text{align}}$ measures the discrepancy between original features and those moved along the dense trajectories. The second term $\mathcal{L}_{\text{smooth}}$ is calculated on the corresponding features before and after a motion supervision step, which is introduced to encourage smooth feature transformations. Finally, $\mathcal{L}_{\text{mask}}$ calculates the contextual deviations of latent feature maps within the target mask, helping to maintain the homogeneity of the latent feature maps. $\beta$ is another task-specific parameter while $\lambda$ is set the same value for all tasks. The detailed definitions of these three terms are as follows:

$$\mathcal{L}_{\text{align}} = \left\| f_{\mathcal{G}^{l+1}}(\hat{\mathbf{z}}_t^{n+1}) - f_{\mathcal{G}^0}(\mathbf{z}_t) \right\|_1, \tag{10}$$

where $f_{\mathcal{G}}(\mathbf{z})$ denotes the latent features of dense points in $\mathcal{G}$, retrieved from $\mathbf{z}$ via feature interpolation, e.g., $f_{\mathcal{G}^{l+1}}(\hat{\mathbf{z}}_t^{n+1})$ means the dense point features at the $l + 1$ position along the trajectories after $n + 1$ optimization steps. Similarly,

$$\mathcal{L}_{\text{smooth}} = \left\| f_{\mathcal{G}^{l+1}}(\hat{\mathbf{z}}_t^{n+1}) - f_{\mathcal{G}^l}(\hat{\mathbf{z}}_t^n)) \right\|_1, \tag{11}$$

and,

$$\mathcal{L}_{\text{mask}} = \left\| (\hat{\mathbf{z}}_t^{n+1} - \mathbf{z}_t) \odot (1 - \mathcal{M}) \right\|_1. \tag{12}$$

In practice, we find that including all intermediate features in the denoising UNet to optimize Eq. (9) leads to better performance.

**Task-aware point tracking**. At last, we update the positions of the dense points after every $N$ motion supervision steps. Thanks to the reliability of our task-aware trajectories, for each point in $\mathcal{P}^*$, we determine its new position as the one on its corresponding trajectory that minimizes Eq. (10), rather than searching the neighboring areas of the trajectories. This allows our handle points to move along the trajectories and avoid undesirable drifts.

## 4 EXPERIMENTS

In this section, we provide quantitative and qualitative analysis of GDrag. Due to the page limitation, more experimental results can be found in the Appendix.

### 4.1 SETUP

**Implementation details**. All our experiments are conducted with an NVIDIA GeForce RTX 4090 graphical card (24 GB). We use Stable Diffusion 1.5 (Rombach et al., 2022) as the base diffusion model. Following (Liu et al., 2024), to ensure the consistency of image content before and after editing, we perform LoRA fine-tuning on the input image, with 80 fine-tuning steps and the adaptor rank is 16. SMS uses the Adam optimizer with a learning rate of 0.01 for $\mathbf{b}$ and 0.02 for $\mathbf{s}$, and the parameter $\lambda$ is set to 0.2. In addition, the values of parameters $\mathcal{L}$, $\mathcal{T}$, and $\mathcal{N}$ are 50, 35, and 5, respectively. We set $r = 5$ and $k = |\mathcal{M}|/10$ to calculate the boundary band in the non-rigid transformation task. In the relocation and in-plane rotation tasks, the value of $\rho$ is set to 1.0, while in the non-rigid transformation and out-of-plane rotation tasks, the value of $\rho$ is 0.2. We calculate the value of $\beta$ adaptively based on the specific task and the details can be found in Appendix A.10.

**Baselines and evaluation metrics**. We compare GDrag with state-of-the-art methods on the Drag-Bench dataset (Shi et al., 2024b) to validate its effectiveness. We select four baselines with publicly available implementations for comparison, including DragDiffusion (Shi et al., 2024b), DragonDiffusion (Mou et al., 2024), FreeDrag (Ling et al., 2024), and DragNoise (Liu et al., 2024). Following these methods, we use LPIPS (Zhang et al., 2018) and mean distance (MD) as our evaluation metrics. LPIPS is widely used to access image quality. The second metric is the average of the Euclidean distances between the target and actual positions of handle points, which reflects how well the edited results align with the user intents.

### 4.2 COMPARISON WITH STATE-OF-THE-ARTS

**Quantitative analysis**. Table 2 reports the performance of GDrag and the other baseline methods. GDrag achieves the lowest LPIPS value (0.0915), which suggests its edited images have the best quality. Furthermore, GDrag obtains the lowest mean distance (26.49), which we owe to our anti-ambiguity dense trajectories.

Beyond these two objective metrics, we also conduct a human study to further evaluate our method. We invite 20 volunteers to complete a questionnaire consisting of 30 questions. We show a group of five edited images produced by GDrag and the four baselines in each question, and ask the volunteers to select the most satisfying one. The results in Table 2 show that most volunteers (about 60.33%) choose GDrag, indicating the significant advantages of GDrag.

Table 2: Quantitative comparison with state-of-the-arts on DragBench.

| Method | MD↓ | LPIPS↓ | User Study ↑ |
|---|---|---|---|
| DragDiffusion (Shi et al., 2024b) | 33.91 | 0.0940 | 7.83% |
| DragonDiffusion (Mou et al., 2024) | 31.63 | 0.1033 | 10.5% |
| FreeDrag (Ling et al., 2024) | 27.41 | 0.0996 | 8.67% |
| DragNoise (Liu et al., 2024) | 29.56 | 0.1017 | 12.67% |
| GDrag (Ours) | **26.49** | **0.0915** | **60.33%** |

**Qualitative analysis**. Figure 4 shows several visual examples generated by GDrag and the baseline methods. These results show that the baselines still produce noticeable artifacts or fail to meet the editing requirements. For instance, in the rotation task, DragDiffusion and FreeDrag struggle with large out-of-plane rotations, while DragonDiffusion and DragNoise fail to preserve the targets effectively. Even in the seemingly simple relocation task, some targets moved by the baselines lag behind the handle points. These examples support our view that implicitly representing various editing tasks leads to severe ambiguities and performance degradation. On the contrary, GDrag explicitly models the editing tasks and generates more appealing images.

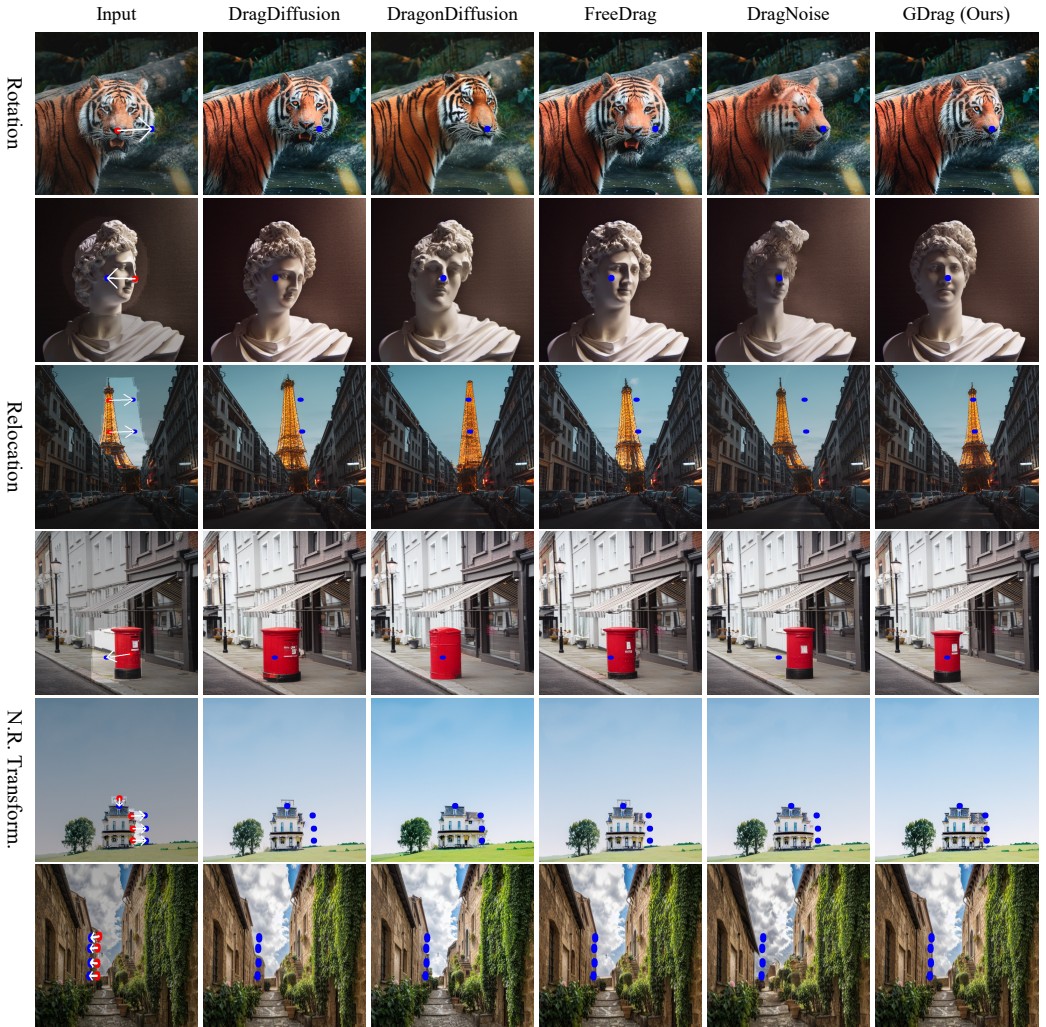

Figure 4: Visual comparison between GDrag and state-of-the-art methods. Blue dots denote the final positions of handle points. GDrag achieves more precise manipulations and satisfying results across various editing tasks, including rotation, relocation, and non-rigid transformation.

## 4.3 ABLATION STUDIES

We conduct ablation studies on the two key components of our framework, i.e., ADT and SMS.

**Effects of anti-ambiguity dense trajectory**. As shown in Table 3, the mean distance metric of the baseline decreased by approximately 20.11%, when augmented by the proposed ADT method. This indicates that incorporating the task prior and rich context (including semantic and geometric neighbors) can significantly improve the precision of manipulations.

Table 3: Effects of the proposed components.

| ADT | SMS | MD↓ | LPIPS↓ |
|-----|-----|-----|--------|
|     |     | 33.91 | 0.0940 |
| ✓   |     | 27.09 | 0.0948 |
| ✓   | ✓   | 26.49±0.07 | 0.0915±7.33e-7 |

**Effects of self-adaptive motion supervision**. Table 3 shows the performance of the baseline can be further improved by the SMS method consistently in both the mean distance and LPIPS metrics. This is reasonable, as the optimizable latent features biases and scaling maps better represent local and fine-grained modifications. Fur-

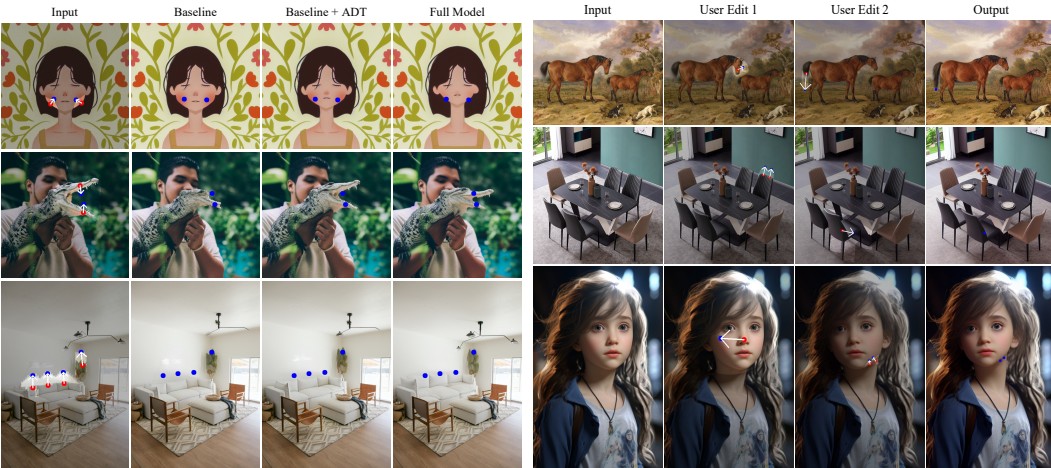

Figure 5: Visual ablation study on the proposed components.

Figure 6: Visual examples of combining atomic manipulations to achieve complex editing.

thermore, the task-aware trajectories and parameters ($\rho$ and $\beta$) help to reduce the difficulty of optimizations. Figure 5 also suggests that with both ADT and SMS, our method obtains the most appealing results.

Moreover, since we randomly select denoising time steps for optimizations in SMS, we report the standard deviations w.r.t. the two evaluation metrics across five experiments with different random seeds. As shown in Tables 3 & 8, and Figure 15, the overall performance of GDrag is stable and hence our random sampling strategy is considerable.

**Compositional editing**. Figure 6 shows that our defined atomic manipulation tasks can be combined to accomplish complex intents. This validates that GDrag is a general-purpose solution for interactive point-based image editing.

## 5    CONCLUSION

In this paper, we present GDrag, a general-purpose optimization-based framework that can tackle diverse interactive point-based image editing tasks. In contrast to conventional methods, GDrag defines three atomic editing tasks to calculate task-aware trajectories and reduce ambiguities. Furthermore, by introducing the self-adaptive motion supervision strategy, which contains optimizable biases and scaling maps for randomly sampled latent features at arbitrary denoising steps, our method better balances content consistency and editing flexibility. Our GDrag method is validated thoroughly on the DragBench dataset and achieves state-of-the-art performance in terms of manipulation precision and image quality.

## ACKNOWLEDGEMENTS

This work was supported by National Key Research and Development Program of China(2024YFE0203100), National Natural Science Foundation of China (NSFC) under Grants No.62372482 and No.62476293, Nansha Key R&D Program under Grant No.2022ZD014, and General Embodied AI Center of Sun Yat-sen University.

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

## A  APPENDIX

The outline of the Appendix is as follows:

• More visual results compared with state-of-the-art methods in A.1;

• User study on intention understanding in A.2;

• Effects of segmentation models in A.3;

• Computational cost analysis of the proposed method in A.4;

• Ablation study on the proposed anti-ambiguity dense trajectory (ADT) calculation method in A.5;

• Ablation study on the proposed self-adaptive model supervision (SMS) strategy in A.6;

• Comparison between using random and fixed denoising time steps in A.7;

• Edited images with different random seeds in A.8;

• An example of our graphical user interface in A.9;

• Our strategy of calculating the task-related parameter $\beta$ in A.10;

• Ablation Study on the task-aware parameters $\rho$ and $\beta$ in A.11.

### A.1  MORE VISUAL RESULTS

Here we provide more qualitative results of GDrag in Figure 9, which further demonstrate that GDrag outperforms state-of-the-art methods across various editing tasks, including rotation, relocation, and non-rigid transformation.

In Figure 7, we show examples of the proposed GDrag method in completing complex manipulations. Each of these manipulations involves motions of multiple joints/parts and incorporates more than one atomic task. For example, in the first row, our goal is to transform a barking dog into a smiling one, which requires us to first close its mouth and then lift the corners of its lips. In these examples, we separate each manipulation into two steps and show the intermediate and final edited images. These results demonstrate that the edited images generated by our GDrag method better align with user intentions and have fewer artifacts compared with the baseline.

In Figure 8, we demonstrate a few failure cases of the proposed method. Although GDrag moves handle points to their target positions more accurately compared with other methods, it generates a few realistic but unnecessary details around the targets, e.g., the leaves on the left side of the rose in the bottom row of Figure 8. These failure cases are mainly caused by the limited inpainting ability of our base generator (Stable Diffusion 1.5), which can be addressed by more advanced generators.

### A.2  USER INTENTION UNDERSTANDING

One of our key assumptions in this paper is that 2D dragging inevitably causes intention ambiguity, which can be verified on two levels: (i) First, given a dragging-based manipulation, will most users recognize it as the same editing task? (ii) If so, can this editing task be recognized using current methods?

To answer these two questions, we conduct an additional user study. In this study, we invite 20 volunteers to complete a questionnaire consisting of 20 questions. In each question, we present a sample from the DragBench dataset and ask the respondents to identify the corresponding user intention by selecting one of the following four choices: (a) Relocation, (b) Rotation, (c) Non-rigid transformation, and (d) Not sure, may be ambiguous.

To evaluate the consistency of the respondents' votes, we adopt the variation ratio (VR) which is defined as $vr = 1 - \frac{F}{K}$, where $F$ is the maximum frequency for a choice in a question and $K$ is the number of the respondents. The higher the VR, the more dispersed the choices are, indicating that the corresponding user intention is more ambiguous.

The average VR of all samples in our user study is $0.47$, with the VR for each sample varying between $0.3$ and $0.7$. Figure 10 also shows some samples of the study. These results validate

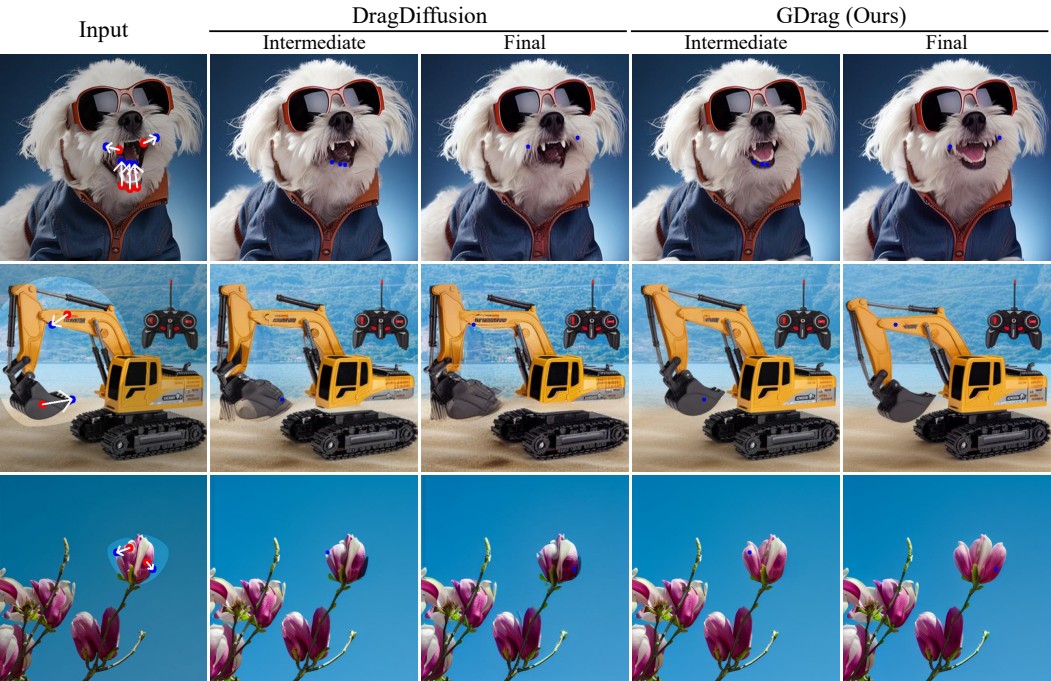

Figure 7: Visual examples of the proposed method in addressing complex motions.

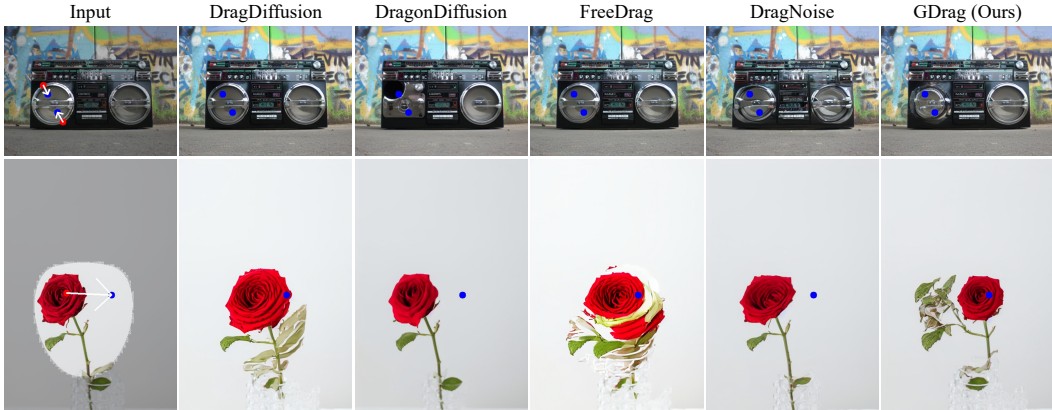

Figure 8: Visualization of failure cases.

that even for the same 2D dragging manipulation, different people have varied interpretations of its intention and are not always consistent. Therefore, the concept of our method, which represents user intentions through atomic tasks, is necessary.

We then investigate whether user intentions can be predicted. Impressed by the powerful reasoning abilities of modern multi-modal large language models (MLLMs), we propose to use them to interpret user intentions. We use images with drawn dragging trajectories and the following prompt for MLLM reasoning: "The white arrow in the figure represents the drag editing starting from the red dot to the blue dot that the user needs to implement. Please select which type of task the user is most likely to want to implement and why, from the following three task types: relocation, rotation, or non-rigid transformations."

We use Qwen2.5, one of the most advanced MLLMs that is open and free, to complete the aforementioned questionnaire as well. We find that only $40\%$ of its choices are consistent with those selected by most respondents. Figure 10 demonstrates some of its results, we can see that even for

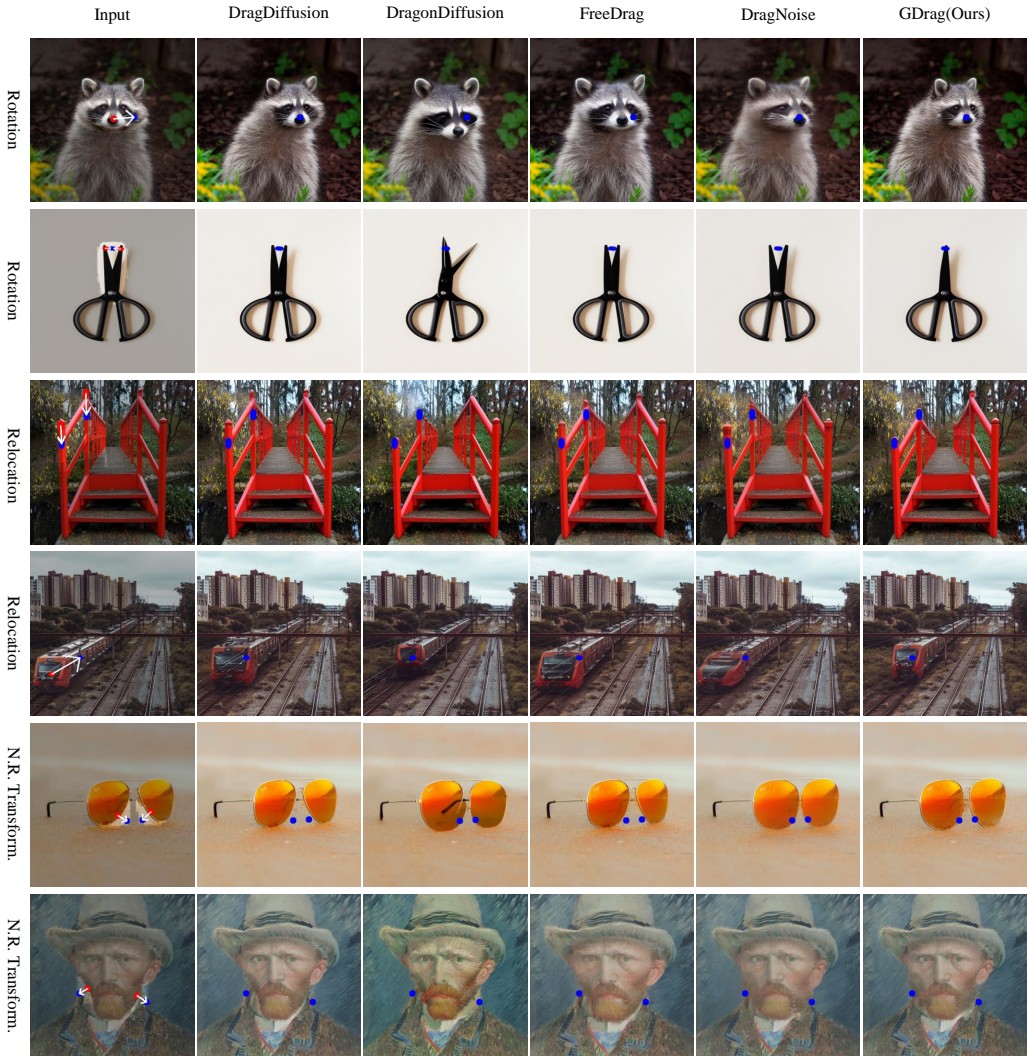

Figure 9: More visual comparison between GDrag and other state-of-the-art methods.

a less ambiguous sample like the rotation task in the first row, the MLLM mistakenly considers it as the non-rigid transformation. With these results, we believe that it is still challenging for current methods to understand user intentions given dragging trajectories. Consequently, our method that explicitly specifies atomic tasks is effective and efficient.

## A.3 EFFECTS OF SEGMENTATION MODELS

Our GDrag utilizes semantic segmentation to calculate dense trajectories. To investigate how the segmentation results affect the performance of GDrag, except for SAM (Kirillov et al., 2023), we also employ Mobile-SAM (Zhang et al., 2023a) for evaluation, which is the light-weight version of SAM.

Considering that MobileSAM uses only 1.5% of the parameters of SAM, it is reasonable

Table 4: Ablation study on the segmentation model.

| Method | MD↓ | LPIPS↓ |
|---|---|---|
| GDrag w/ MobileSAM | 26.74 | 0.0959 |
| GDrag w/ SAM | 26.49 | 0.0915 |

that the masks predicted by MobileSAM are less accurate. However, from the quantitative results in Table 4, we observe that GDrag still performs well with MobileSAM. The mean distance and LPIPS

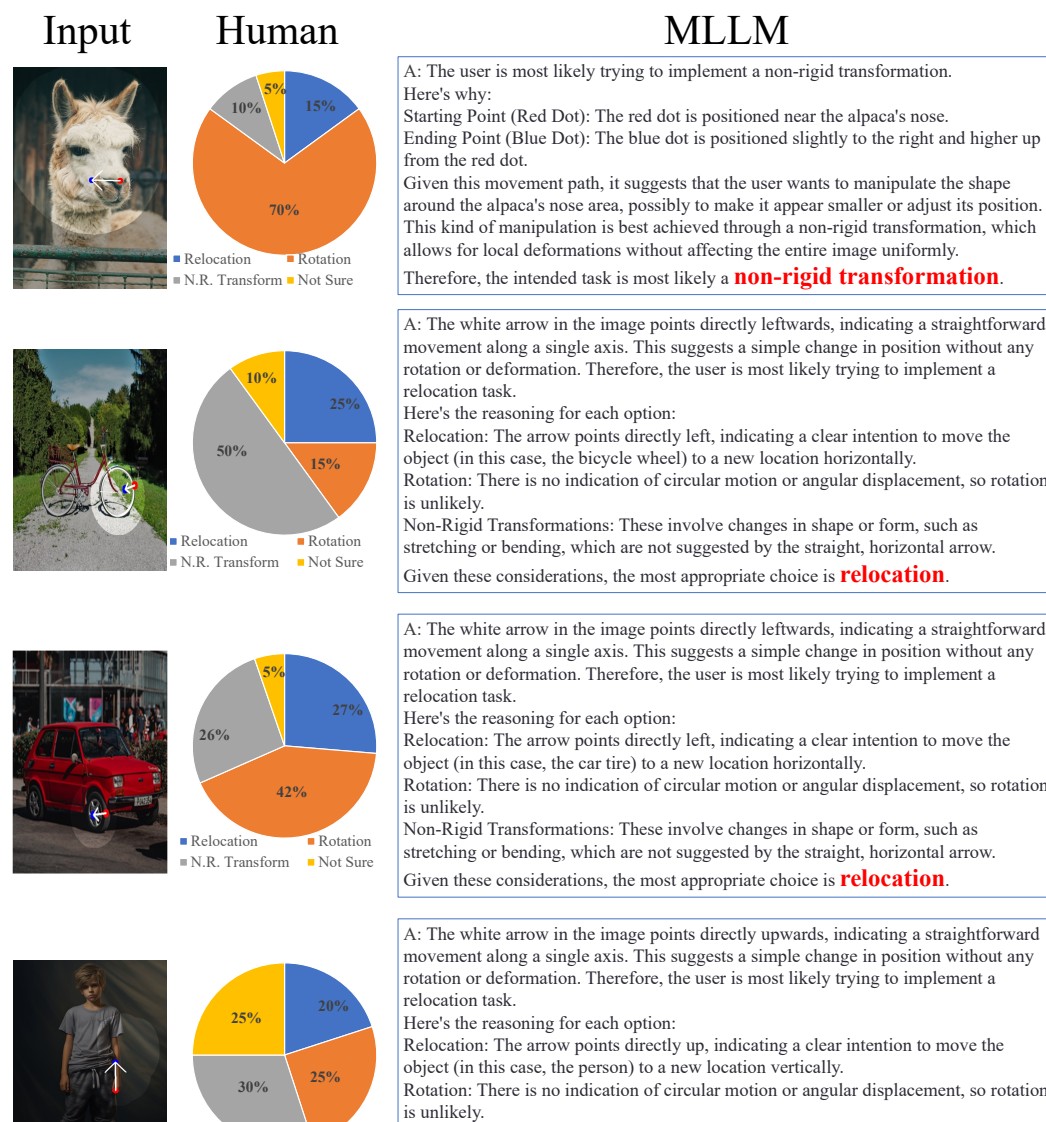

Figure 10: Visual samples of the study on understanding user intentions.

of GDrag with MobileSAM are 26.74 and 0.0959, respectively, while those of GDrag with SAM are 26.49 and 0.0915. Figure 11 also shows that, despite some artifacts like holes and disconnected regions in the predicted masks, GDrag still generates high-quality edited images.

## A.4  COMPUTATIONAL COST ANALYSIS

Our computational cost analysis is conducted using an NVIDIA GeForce RTX 4090 GPU with 24 GB of memory. We calculate the mean processing time per image and the GPU memory consumption on the DragBench dataset (Shi et al., 2024b), with a fixed image size of $512 \times 512$ and 250 optimization steps. We select DragDiffusion (Shi et al., 2024b) as the baseline for comparison as it also adopts Stable Diffusion 1.5 as the base generator. The computational cost of the baseline is approximately 12 GB of memory and 132 seconds per image, including 40 seconds for employing low-rank adaption (LoRA) and 92 seconds for optimization. In comparison, GDrag consumes about 16 GB of memory and 192 seconds per image, including 40 seconds for LoRA, 2 seconds

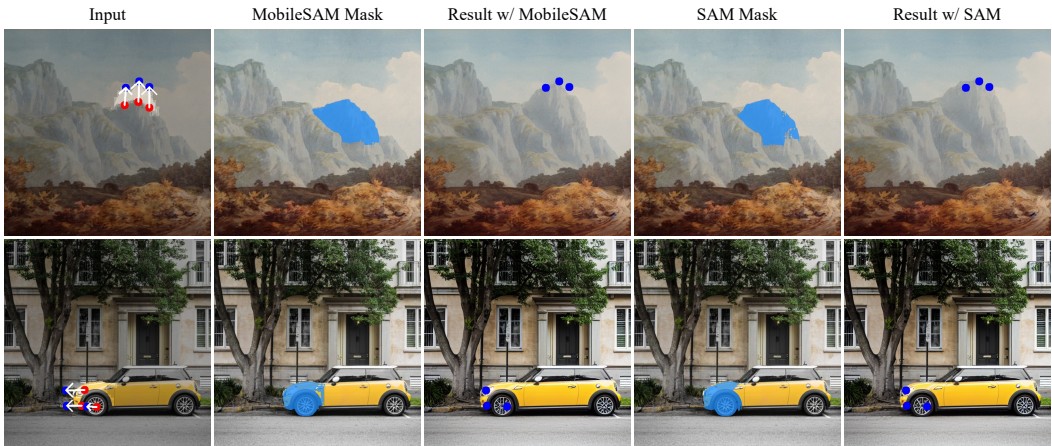

| Input | MobileSAM Mask | Result w/ MobileSAM | SAM Mask | Result w/ SAM |

Figure 11: Visual comparison of the proposed method with different segmentation models.

for segmentation, and 150 seconds for optimization. We find that the additional computational cost is proportional to the task difficulty. For easier tasks like relocation and in-plane rotation, the processing time of GDrag is almost the same as the baseline, while more complex manipulations like out-of-plane rotation and non-rigid transformation take longer. Considering that our task prioritizes image quality over processing speed and that most current methods are not real-time, the proposed GDrag remains competitive.

### A.5 ABLATION STUDY ON ADT

The proposed ADT method converts sparse handle points specified by users into task-aware dense trajectories to better exploit contextual information. To ensure the fairness of comparison and validate that the advantages of the proposed method do not merely rely on more handle points, we conduct an ablation study that replaces the inputs of the baselines with dense trajectories. The results reported in Table 5 show that not all baselines benefit from dense trajectories. Although DragDiffusion (Shi et al., 2024b) and DragonDiffusion (Mou et al., 2024) with dense trajectories achieve lower mean distances, the performance of Free-Drag (Ling et al., 2024) and DragNoise (Liu et al., 2024) deteriorates. This is because Free-Drag devises its own strategy for trajectory cal-

Table 5: Quantitative comparison of different methods with ADT.

| Method | MD↓ | LPIPS↓ |
|---|---|---|
| DragDiffusion | 33.91 | 0.0940 |
| DragDiffusion w/ ADT | 33.48 | 0.1559 |
| DragonDiffusion | 31.63 | 0.1033 |
| DragonDiffusion w/ ADT | 29.16 | 0.1109 |
| FreeDrag | 27.41 | 0.0996 |
| FreeDrag w/ ADT | 35.16 | 0.1061 |
| DragNoise | 29.56 | 0.1017 |
| DragNoise w/ ADT | 30.00 | 0.1399 |
| GDrag (Ours) | **26.49** | **0.0915** |

culation, while DragNoise conducts high-level feature optimizations and already adopts whole segmentation masks for guidance. Therefore, we can conclude that the performance gains of GDrag mainly stem from its method, instead of using more handle points.

This can also be validated by the visual examples shown in Figure 12. In these examples, the end positions of handle points of the baselines with dense trajectories are closer to the targets. However, their results still have noticeable artifacts compared with the proposed method, such as the warped fire extinguishers in the last example.

### A.6 ABLATION STUDY ON SMS

Our SMS method introduces learnable latent biases and scaling maps to balance the modification and preservation of latents. To further validate its effectiveness, we perform an ablation study considering five variants of SMS, including: (i) "Latent" which optimizes la-

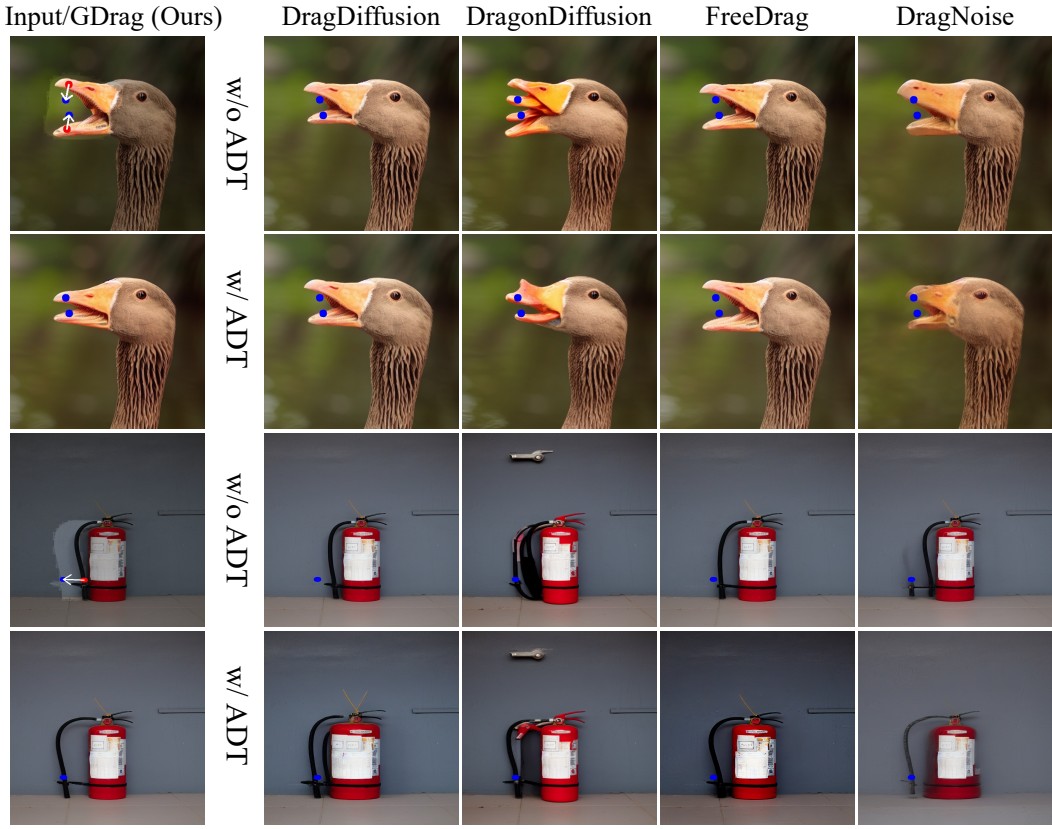

Figure 12: Visual examples of the ablation study on the proposed ADT method.

tent features directly; (ii) "Bias" which only learns latent biases; (iii) "Scaling Scalar" that learns a scaling scalar for the LoRA side branch as in conventional methods; "Scaling Map" that converts the scalar in (iii) to a learnable map; (v) "Bias + Scaling Scalar" that combines (ii) and (iii). We employ the proposed ADT method with all these five variants.

Table 6 reports the quantitative comparison among the variants of SMS. We observe that, compared with optimizing latents directly, "Bias", "Scaling Scalar", and "Bias + Scaling Scalar" reduce the LPIPS values significantly yet perform badly when evaluated by the mean distance metric. This suggests that while these variants tend to preserve the original images and generate high-fidelity images, they are less effective at following user intentions. The "Scaling Map" variant performs better than "Latent" on LPIPS, as it allows us to employ LoRA in a finer manner. Nevertheless, without the optimization of latents or latent biases, this variant is still inferior for diverse modifications and falls behind "Latent" on the mean distance. Finally, with both optimizable components, our SMS method outperforms the latent-based optimization on both metrics consistently.

Table 6: Quantitative comparison of SMS variants.

| Method | MD↓ | LPIPS↓ |
|---|---|---|
| Latent | 27.09 | 0.0940 |
| Bias | 47.11 | 0.0522 |
| Scaling Scalar | 47.92 | **0.0486** |
| Scaling Map | 28.62 | 0.0873 |
| Bias + Scaling Scalar | 47.15 | 0.0516 |
| SMS | **26.49** | 0.0915 |

The above analysis is further validated by the qualitative examples shown in Figure 13. It can be observed that the images synthesized by "Bias", "Scaling Scalar", and "Bias + Scaling Scalar" are almost identical to the input images. The "Latent" and "Scaling Map" variants modify the images to a certain extent but they also result in distorted target structures.

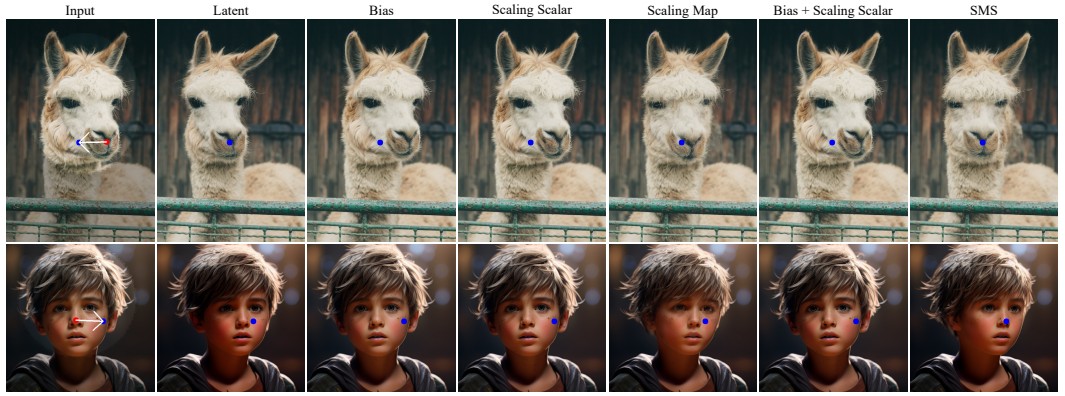

Figure 13: Visual examples of the ablation study on the proposed SMS strategy.

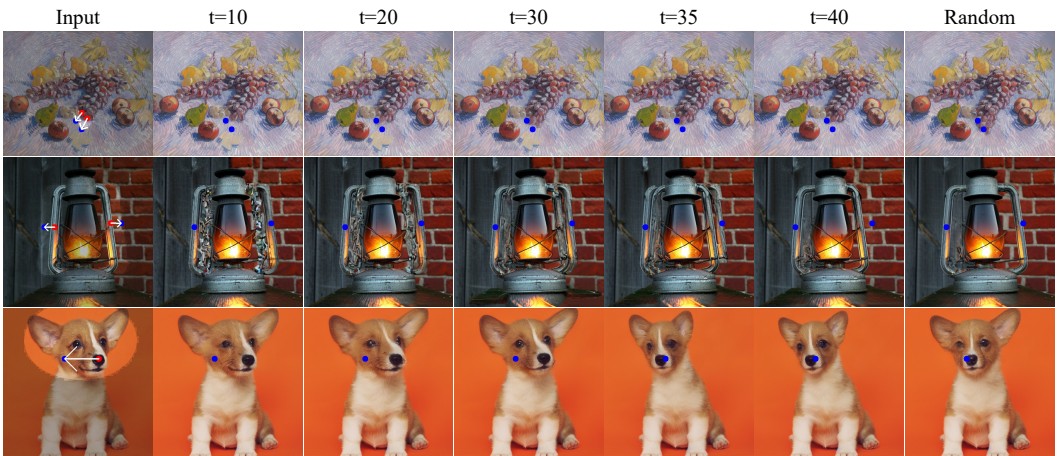

Figure 14: Visual comparison between randomly sampled and fixed denoising steps.

### A.7 OPTIMIZATION RESULTS UNDER DIFFERENT DENOISING STEPS

Another key idea of our SMS method is that previous methods selecting a fixed denoising time step $t$ are insufficient to complete various manipulations. This is because a large $t$ generally results in large structural changes, while a small $t$ is more suitable for small modifications. Hence, we propose sampling time steps randomly during optimization to cover as many variations as possible and reduce the computational cost of exhausting all time steps. To validate this, we compare SMS with a variant that adopts a fixed time step. As the maximum denoising time step in our base model $T = 50$, we report the results of the variant with $t = 10, 20, 30$, and $40$. Both the quantitative comparison in Table 7 and the qualitative comparison in Figure 14 show that our SMS method with random time steps outperforms the fixed one significantly. Hence, our SMS overcomes the limitations of selecting the time step manually and is more practical for real-world applications.

Table 7: Quantitative comparison with different denoising steps.

| Method | MD↓ | LPIPS↓ |
|--------|-----|--------|
| $t = 10$ | 43.72 | 0.0628 |
| $t = 20$ | 35.05 | 0.0739 |
| $t = 30$ | 30.36 | 0.0873 |
| $t = 35$ | 29.37 | 0.0908 |
| $t = 40$ | 31.66 | 0.0928 |
| Random | **26.49** | **0.0915** |

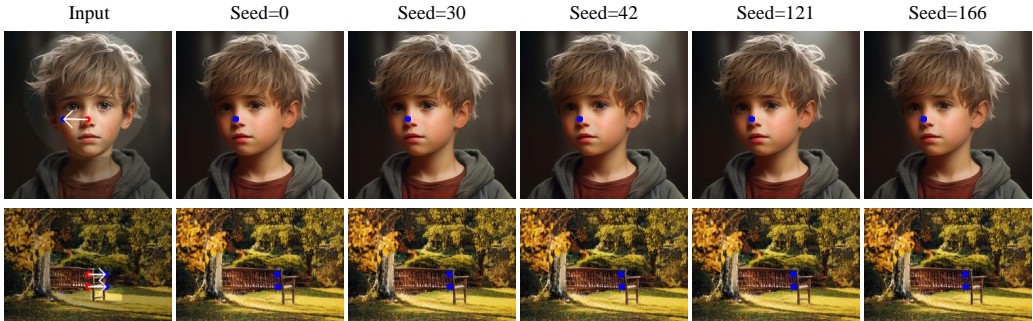

Figure 15: Visual results under different random seeds. These examples are satisfactory and validate that our method is stable.

### A.8 OPTIMIZATION RESULTS UNDER DIFFERENT RANDOM SEEDS

In GDrag, an arbitrary time step is randomly selected for optimization. Therefore, to demonstrate the stability of our method, we visualize the edited images under five different random seeds, as shown in Figure 15. We can see that our method is robust to the choice of random seeds, e.g., in the first row, the rotations of the faces are almost identical, and all results are satisfactory. We also report the quantitative performance of our method with different random seeds in Table 8, which further validates the stability of our method.

Table 8: Quantitative results under different random seeds. MD stands for mean distance.

| Random Seed | MD↓ | LPIPS↓ |
|---|---|---|
| 0 | 26.56 | 0.0907 |
| 30 | 26.25 | 0.0929 |
| 42 | 26.33 | 0.0916 |
| 121 | 26.97 | 0.0919 |
| 166 | 26.33 | 0.0905 |

### A.9 THE GRAPHICAL USER INTERFACE

Figure 16 shows an example of our graphical user interface (GUI), where we guide the user to complete the rotation task. This GUI allows the user to define complex rotations easily, e.g., by dragging an ellipse that roughly covers the editing area and rotating the ellipse to mimic out-of-plane rotations.

### A.10 THE VALUE OF $\beta$ BASED ON DIFFERENT TASKS

Here we will provide a detailed definition of the value of $\beta$ in different tasks:

1) Relocation:

$$\beta^n = \exp(-\frac{n}{N} * \frac{\mathcal{L}_{\text{align}}^{n-1}}{\mathcal{L}_{\text{smooth}}^{n-1}} * \frac{|\mathcal{M}|}{H * W} * 50) * 0.1, \tag{13}$$

where $H$ and $W$ are the height and width of the input image.

2) In-plane Rotation:

$$\beta^n = \exp(-\frac{n}{N} * \frac{\mathcal{L}_{\text{align}}^{n-1}}{\mathcal{L}_{\text{smooth}}^{n-1}} * \frac{|\mathcal{M}|}{H * W}) * 0.7 + 0.3. \tag{14}$$

3) Out-of-plane Rotation:

$$\beta^n = \exp(-\frac{n}{N} * \frac{\mathcal{L}_{\text{align}}^{n-1}}{\mathcal{L}_{\text{smooth}}^{n-1}} * \frac{|\mathcal{M}|}{H * W}). \tag{15}$$

4) Non-rigid Transformation:

$$\beta^n = \exp(-\frac{n}{N} * \frac{\mathcal{L}_{\text{align}}^{n-1}}{\mathcal{L}_{\text{smooth}}^{n-1}} * \frac{|\mathcal{M}|}{H * W} * 25) * 0.2. \tag{16}$$

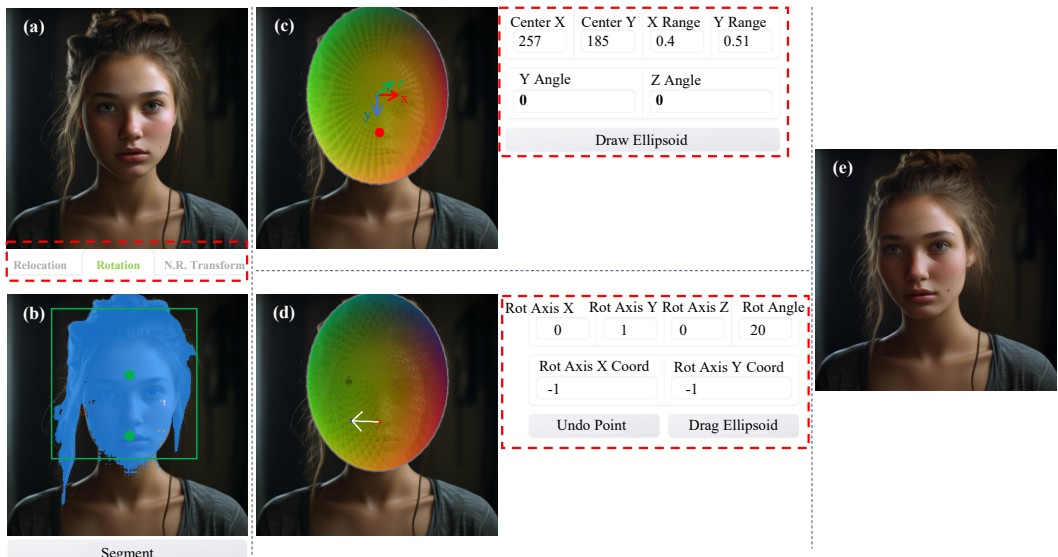

Figure 16: The graphical user interface of rotation task. (a) The user selects the input image and specifies the editing task. (b) The user uses the promptable segmentation model to locate the editing area, with prompts like positive points and bounding boxes. (c) The user drags and customizes an ellipsoid to roughly cover the target, of which parameters like the center coordinates and the lengths of axes are recorded. (d) The user selects an axis and drags it to rotate the ellipsoid, and we record the rotation angle to calculate the corresponding trajectories. (e) The edited image synthesized by our method.

In all tasks, $\beta$ decreases gradually as $n$ increases. This is because more optimization steps will lead to more image information loss at the initial position. Consequently, we need to decrease the value of $\beta$ to increase the weight of $\mathcal{L}_{\text{align}}$ as the step of optimization increases. In order to more balance the weights of $\mathcal{L}_{\text{align}}$ and $\mathcal{L}_{\text{smooth}}$ during the optimization process, we also consider $\mathcal{L}_{\text{align}}$ and $\mathcal{L}_{\text{smooth}}$ in the previous optimization into the formula. The intuition is that if the value of $\mathcal{L}_{\text{smooth}}$ is relatively large in the previous optimization, we should focus more on the $\mathcal{L}_{\text{smooth}}$ in this optimization, which also applies to $\mathcal{L}_{\text{align}}$.

Additionally, for different inputs, we adjust the rate of $\beta$ reduction based on the proportion of the editing area relative to the entire image. Specifically, a larger ratio requires a faster reduction, as the larger the ratio of editing area is, the more easily the image information at the initial position is lost. Finally, for different tasks, the range of $\beta$ values will vary. For example, in the relocation task, $\beta$ needs to be restricted to a relatively small range. This ensures that our model focuses more on $\mathcal{L}_{\text{align}}$, meaning it pays more attention to preserving the initial image information of the edited object during the dragging process. In contrast, the in-plane rotation task requires generating new image information, so $\beta$ needs to be within a larger range.

## A.11 ABLATION STUDY ON TASK-SPECIFIC PARAMETERS

We investigate the effects of our task-related parameters $\rho$ and $\beta$ in the non-rigid transformation task as it is the most challenging among all three atomic tasks. Table 9 reports the results of our method with different values of $\rho$. From the results, we observe that the lowest mean distance is achieved when $\rho = 0.4$ and gradually increases as the value of $\rho$ increases. This is reasonable since the role of $\rho$ is to preserve the original images. Hence, the larger $\rho$ is, the fewer modifications are exhib-

Table 9: Ablation study on $\rho$.

|  | MD↓ | LPIPS↓ |
|---|---|---|
| Baseline | 33.86 | 0.0733 |
| $\rho = 0.2$ | 31.26 | 0.0679 |
| $\rho = 0.4$ | 30.61 | 0.0682 |
| $\rho = 0.6$ | 33.52 | 0.0642 |
| $\rho = 0.8$ | 34.74 | 0.0632 |
| $\rho = 1.0$ | 34.39 | 0.0647 |

ited in the edited images. The LIPIPS scores of $\rho$ with different settings are similar, which suggests that the optimization process of GDrag and the image quality of its outputs are stable.

As for $\beta$, we vary its base scaling factor in Eq. (16) from 0.2 to 1.0 to analyze its effects. The results in Table 10 show that the overall trend of $\beta$ in affecting the performance of our method is similar to that of $\rho$. As discussed in A.10, we use $\beta$ to control the weight of $\mathcal{L}_{\text{align}}$, consequently preserving the initial image information before editing. From the results, we can see that $\beta$ also fulfills our goal, as the mean distance increases (indicating fewer modifications) as the value of $\beta$ increases.

Table 10: Ablation study on $\beta$.

|  | MD$\downarrow$ | LPIPS$\downarrow$ |
|---|---|---|
| Baseline | 33.86 | 0.0733 |
| $\beta = 0.2$ | 30.08 | 0.0663 |
| $\beta = 0.4$ | 30.61 | 0.0682 |
| $\beta = 0.6$ | 33.52 | 0.0642 |
| $\beta = 0.8$ | 34.74 | 0.0632 |
| $\beta = 1.0$ | 34.39 | 0.0647 |

