# OpenReview forum: "GDrag:Towards General-Purpose Interactive Editing with Anti-ambiguity Point Diffusion"
_ICLR.cc/2025/Conference — ICLR 2025 Poster_

### Official Review · Reviewer_jimP · 2024-10-17

**Soundness:** 3
**Presentation:** 3
**Contribution:** 3
**Rating:** 6
**Confidence:** 3

**Summary:**

The primary goal of this work is to address two issues inherent in current point-based image manipulation: intention ambiguity and content ambiguity. To tackle intention ambiguity, the paper defines a taxonomy of atomic manipulations that can be combined to form complex actions. For content ambiguity, the authors introduce the Anti-Ambiguity Dense Trajectory calculation method (ADT) and a Self-Adaptive Motion Supervision method (SMS). In ADT, each atomic manipulation is defined using a dense point set and corresponding point trajectories from an image segmentation model, allowing for better specification of the motion direction for specific tasks. SMS enhances performance by jointly optimizing point-wise adaptation scales and latent feature biases. The proposed methods demonstrate significant advantages in both quantitative and qualitative comparisons.

**Strengths:**

1. The writing is clear, engaging, and easy to understand.
2. The objectives of addressing intention ambiguity and content ambiguity are both reasonable and aligned with practical needs.
3. The process of decomposing drag-based manipulation into three atomic manipulations is concise and effective, facilitating future research.
3. The approach of optimizing $z_0$ to replace optimizing $z_t$ demonstrates a degree of novelty.

**Weaknesses:**

My primary concern is whether the estimation of dense points and point trajectories proposed in the paper will be affected by the performance of the semantic segmentation model, and whether there might be significant deviations for more complex motions, such as the transition of a hand from an open to a closed position.

**Questions:**

Including a discussion of some failure examples would make the paper more comprehensive.

---

> ### Author Response · Authors · 2024-11-24
> **Reply to Reviewer jimP**
>
> Thank you for your high affirmations on our method. We have included the ablation study on segmentation models (A.3) and examples of complex motions (A.1) in our revised paper.
>
> **Q1 Effects of segmentation models**: Our GDrag utilizes semantic segmentation to calculate dense trajectories. To investigate how the segmentation results affect the performance of GDrag, except for SAM  (Kir-
> illov et al., 2023), we also employ MobileSAM (Zhang et al., 2023a)  for evaluation, which is the light-weight version of SAM.
>
> Considering that MobileSAM uses only $1.5\%$ of the parameters of SAM, it is reasonable that the masks predicted by MobileSAM are less accurate. However, from the quantitative results in Table 4, we observe that GDrag still performs well with MobileSAM. The mean distance and LPIPS of GDrag with MobileSAM are $26.74$ and $0.0959$, respectively, while those of GDrag with SAM are $26.49$ and $0.0915$. Figure 11 also shows that, despite some artifacts like holes and disconnected regions in the predicted masks, GDrag still generates high-quality edited images.
>
> **Q2 More complex motion**: This is an interesting question. In fact, we consider generating the transition of a hand from an open to a closed position challenging, not because the motion is complex, but because our base generator (SD1.5) is hard to generate proper hands. We find many studies try to enhance diffusion models to generate hand images [1-3]. Unfortunately, they require extra model training that we cannot afford.
>
> However, we managed to find more common **multi-joints/parts targets that have similar and complex open and closed motions** like hands, such as excavators and flower buds. We provide a qualitative comparison with these objects in A.1 and as follows:
> In Figure 7, we show examples of the proposed GDrag method in completing complex manipulations. Each of these manipulations involves motions of multiple joints/parts and incorporates more than one atomic task. For example, in the first row, our goal is to transform a barking dog into a smiling one, which requires us to first close its mouth and then lift the corners of its lips. In these examples, we separate each manipulation into two steps and show the intermediate and final edited images. These results demonstrate that the edited images generated by our GDrag method better align with user intentions and have fewer artifacts compared with the baseline.
>
> [1] Lu, Wenquan, et al. "Handrefiner: Refining malformed hands in generated images by diffusion-based conditional inpainting." Proceedings of the 32nd ACM International Conference on Multimedia. 2024.
>
> [2] Wang, Chengrui, et al. "RHanDS: Refining Malformed Hands for Generated Images with Decoupled Structure and Style Guidance." arXiv preprint arXiv:2404.13984 (2024).
>
> [3] Pelykh, Anton, Ozge Mercanoglu Sincan, and Richard Bowden. "Giving a Hand to Diffusion Models: a Two-Stage Approach to Improving Conditional Human Image Generation." arXiv preprint arXiv:2403.10731 (2024).

---

> > ### Comment · Reviewer_jimP · 2024-11-26
> >
> > Thank you for the author's response, which has resolved most of the issues. I am willing to maintain my score.

---

### Official Review · Reviewer_DgdH · 2024-10-28

**Soundness:** 2
**Presentation:** 3
**Contribution:** 2
**Rating:** 6
**Confidence:** 4

**Summary:**

This paper proposes a system referred to as GDrag to resolve intention based on user given drag. GDrag introduces two strategies to mitigate content ambiguity, including an anti-ambiguity dense trajectory calculation method and a self-adaptive motion supervision method.

**Strengths:**

[1] Proposed problem (intention-awareness) for drag-based editing is challenging issue.

[2] Proposed method is reasonable

[3] Figure is illustrative

**Weaknesses:**

[1] Introduction: The problem statement lacks persuasiveness due to unclear writing. Begin by explaining what "handle points" in drag-based diffusion are. Don’t assume that all readers are already familiar with the task of drag-based diffusion and the terminology used. Without an explanation of handle points, readers may struggle to understand the trajectories derived from them, which ultimately undermines the intent behind the examples illustrating the problem.

[2] Intention Understanding Model: Although the paper is presented as a model for understanding intention, the method proposed seems more focused on effectively executing predefined intentions rather than genuinely understanding them.

[3] Practicality: Since the approach relies on heuristic methods based on categorical intentions, its practical applicability appears limited.

[4] Computational Analysis: Functionally, placing dense points is likely to lead to significant computational overhead. However, there is a lack of experimental validation, as no computational analysis has been conducted to assess this aspect.

**Questions:**

My questions are based on weakness. Please give me a rebuttal on them.
User intention may highly differ from drag, is there any analysis of understanding human intention based on input drag?

---

> ### Comment · Reviewer_DgdH · 2024-11-25
>
> I read authors' rebuttal and they address my concerns well, I increase my score. Thank you!

---

### Official Review · Reviewer_wdkj · 2024-11-03

**Soundness:** 3
**Presentation:** 3
**Contribution:** 3
**Rating:** 8
**Confidence:** 3

**Summary:**

The paper introduces GDrag, a novel task-aware, optimization-based framework designed for interactive image editing. This method addresses the limitations of existing point-based diffusion models, particularly the challenges of intention ambiguity and content ambiguity in image editing tasks. Existing point-based methods, such as DragDiffusion and FreeDrag, struggle with accurately modeling diverse editing tasks, often leading to mixed or unclear trajectories (intention ambiguity) and a lack of precise target identification (content ambiguity). Current approaches also face challenges in representing 3D manipulations, relying too much on single denoising time steps.

 To overcome these issues, GDrag introduces three atomic editing tasks—relocation, rotation (both in-plane and out-of-plane), and non-rigid transformations (like scaling or content creation/removal). This task-aware design allows the system to simplify complex manipulations by breaking them into smaller, specific tasks. ADT (Anti-ambiguity Dense Trajectory Estimation): This component improves the precision of 3D edits by selecting the semantic and geometric neighbors of handle points, allowing the creation of dense, contextually informed trajectories rather than simple 2D lines. SMS optimizes latent features by sampling them from various denoising steps, enabling a more detailed control of motion and addressing content ambiguity. Additionally, SMS applies low-rank adaptation techniques, allowing GDrag to preserve target details at multiple granular levels.

**Strengths:**

## Originality
The paper identifies and addresses key limitations in existing point-based image editing methods, specifically intention ambiguity and content ambiguity, offering a well-defined solution to these issues.
- By categorizing editing actions into distinct tasks (relocation, rotation, non-rigid transformations), GDrag allows for more targeted and effective edits. This task-oriented approach enables clearer control over image transformations and reduces the complexity of handling diverse editing requirements.
- Innovative Dense Trajectory Estimation (ADT): ADT is a significant advancement in managing trajectory information. It enhances the precision and reliability of editing by creating a dense point set with contextual information, which is especially valuable for complex 3D manipulations
- Self-Adaptive Motion Supervision (SMS): The SMS method introduces a robust way to optimize latent features across multiple denoising steps. This enhances generative models’ performance by allowing finer-grained control and preserving target details at various levels, improving the overall quality of the edits.

## Quality
GDrag is evaluated on DragBench, a benchmark that effectively demonstrates its advantages over existing methods. The results show both quantitative and qualitative improvements in trajectory accuracy and image quality, providing strong empirical evidence for its effectiveness. Compared to other baseline models, GDrag consistently delivers more precise and visually appealing edits, highlighting its promising performance in the interactive image editing.

## Clarity
The overall organization and writing of the paper are well-executed. It clearly articulates the significant limitations of current methods in point-based image manipulation task and presents an effective solution. The logic and details of the proposed method are well thought out, with no apparent flaws.

## Significance
I believe the main work of this paper makes a substantial contribution to the task, particularly in defining the issues of intention and content ambiguity and proposing an effective solution. The overall approach is highly intuitive and, in my opinion, offers valuable insights that could inspire future research in this area.

**Weaknesses:**

- I personally think that adding more details about the core modules in the ablation study section could make it easier for readers to follow and better grasp the key aspects of these modules.
- In lines 299-302, the authors employ a random optimization step. I suggest that adding a comparison between fixed and variable optimization steps would enhance the persuasiveness of the paper, providing a clearer understanding of the benefits of using a variable approach.

**Questions:**

- A detailed definition of the Distance metric would be beneficial, particularly in distinguishing it from the Mean Distance metric used in DragDiffusion [1]. Providing a more comprehensive explanation of the Distance metric, including its calculation and interpretation, could help clarify its role in evaluating model performance. Supplementing these details in the Appendix would enhance readability for new readers, offering them a clearer understanding.



[1] DragDiffusion: Harnessing Diffusion Models for Interactive Point-based Image Editing. Shi et al.

**Details Of Ethics Concerns:**

- Data Privacy and Consent: If the paper uses or references datasets involving real individuals or private images, a review for data privacy and consent would be important.

- Potential for Misuse: Image editing techniques, especially those that allow realistic manipulations, may have implications for misinformation, deepfake creation, or altering real-life identities.

- Transparency and Disclosure: If the methods are intended to generate or modify images in a way that might deceive users or viewers without disclosure, ethical implications could arise.

---

### Official Review · Reviewer_1gRT · 2024-11-03

**Soundness:** 3
**Presentation:** 3
**Contribution:** 2
**Rating:** 6
**Confidence:** 4

**Summary:**

The paper introduces a task-aware, optimization-based framework for general-purpose interactive editing (GDrag).  GDrag categorizes point-based image manipulations into three atomic tasks based on user intents, and convert sparse trajectories into dense trajectories via a carefully designed graphical user interface. Based on the converted dense trajectories, a set of fine-grained optimization parameters are applied for motion supervision. GDrag achieves superior performance on DragBench.

**Strengths:**

1. GDrag focuses on solving the ambiguity of user intents. The proposed Anti-ambiguity Dense Trajectory (ADT) Estimation, i.e., the graphical user interface is interesting, especially for rotation task.
2. The paper is easy to understand.

**Weaknesses:**

1. There are not enough experiments to prove the effectiveness of the proposed Self-adaptive Motion Supervision (SMS) module.
2. The definition of symbols is confusing.
3. The experiments are not enough to show the superiority of the propose method.

**Questions:**

1. The ablation study in Table 3 shows that the proposed Self-adaptive Motion Supervision (SMS) barely contributes to the performance improvements. It will be better to provide comparison between traditional motion supervision and SMS.
2. Since GDrag uses augmented dense trajectories for motion supervision, the qualitative comparison among other state-of-the-art methods seems a little bit unfair. What do the results of these methods look like using denser trajectories?
3. The computational complexity in line 299 seems not right. The compared methods usually select a fixed timestep and optimize N times, which is much less than O(TL).
4. Is it necessary to use features of the original UNet? Why not use features of UNet w/ LoRA directly?
5. Is there any experiments on the variation of hyper-paramters \rho and \beta?

---

### Official Review · Reviewer_54bB · 2024-11-04

**Soundness:** 3
**Presentation:** 3
**Contribution:** 3
**Rating:** 6
**Confidence:** 5

**Summary:**

This paper proposes GDrag, a general-purpose optimization-based framework to tackle diverse interactive point-based image editing tasks. GDrag introduces two strategies, an anti-ambiguity dense trajectory calculation method (ADT) to calculate the trajectories, and a self-adaptive motion supervision method (SMS) to refine latent features. The experiment performance demonstrates the powerful editing capabilities of GDrag.

**Strengths:**

1.	Originality: The paper addresses the issue of intent ambiguity in dragging-based image editing tasks with an anti-ambiguity dense trajectory calculation method, and specifically constrains movement on dense trajectories, demonstrating good originality overall.
2.	Quality: The paper clearly outlines its innovations, presents a reasonable discussion, provides sufficient experimental results, and demonstrates good quality.
3.	Clarity: The discussion is mostly clear.
4.	Significance: The method proposed in this paper facilitates users in expressing their intentions, enhances interaction, and achieves competitive results, making it of considerable significance.

**Weaknesses:**

The clarity could be improved. For instance, the term fP* mentioned in lines 345-346 does not appear in Equation 10.

**Questions:**

1.	The "rotation" in the middle row of Figure 3 is somewhat unclear. Is it intended to rotate the back edge of the bottle to the front? Please provide a clearer explanation here.
2.	The reasoning behind the use of low-rank adaptation in line 324 is somewhat vague. Please provide a more detailed explanation here.

---

### Meta-Review · Area_Chair_tqvE · 2024-12-16

**Metareview:**

This paper proposes a task-aware, optimisation-based framework designed for interactive point-based image editing. It addresses intention ambiguity by defining atomic manipulation tasks and mitigates content ambiguity through two key strategies: Anti-Ambiguity Dense Trajectory (ADT) calculation, which refines motion trajectories using semantic and geometric context, and Self-Adaptive Motion Supervision (SMS), which optimises latent features for precise control.  All reviewers agree that the paper is well-structured and clearly identifies the limitations of existing point-based image editing methods. They also agree that the proposed framework is intuitive, well-detailed, and demonstrates strong experimental performance.

**Additional Comments On Reviewer Discussion:**

Some reviewers raised concerns about the clarity of certain parts of the paper, including ambiguous notations and a lack of failure examples, which were addressed during the rebuttal. Reviewer 1gRT highlighted three specific issues: insufficient experiments to demonstrate the effectiveness of the SMS module, unclear computational complexity, and missing experiments on hyper-parameter. However, they did not provide post-rebuttal comments, and the AC believes the rebuttal adequately addressed these concerns. Reviewer DgdH expressed concerns that the method seemed more focused on executing predefined intentions than understanding them and questioned the practical applicability of heuristic methods based on categorical intentions. Following the rebuttal, which addressed these points, the reviewer raised their score to "marginally above the acceptance threshold." Overall, all reviewers have expressed a positive outlook on this work.

---

### Decision · Program_Chairs · 2025-01-22

Accept (Poster)